# The Effects of Intermittent Fasting on Brain and Cognitive Function

**DOI:** 10.3390/nu13093166

**Published:** 2021-09-10

**Authors:** Jip Gudden, Alejandro Arias Vasquez, Mirjam Bloemendaal

**Affiliations:** 1Department of Psychiatry, Donders Institute for Brain, Cognition and Behaviour, Radboud University Medical Center, 6525 GA Nijmegen, The Netherlands; jipgudden@hotmail.com (J.G.); alejandro.ariasvasquez@radboudumc.nl (A.A.V.); 2Department of Human Genetics, Donders Institute for Brain, Cognition and Behaviour, Radboud University Medical Center, 6525 GA Nijmegen, The Netherlands

**Keywords:** intermittent fasting, cognition, brain-related diseases, prevention and progress

## Abstract

The importance of diet and the gut-brain axis for brain health and cognitive function is increasingly acknowledged. Dietary interventions are tested for their potential to prevent and/or treat brain disorders. Intermittent fasting (IF), the abstinence or strong limitation of calories for 12 to 48 h, alternated with periods of regular food intake, has shown promising results on neurobiological health in animal models. In this review article, we discuss the potential benefits of IF on cognitive function and the possible effects on the prevention and progress of brain-related disorders in animals and humans. We do so by summarizing the effects of IF which through metabolic, cellular, and circadian mechanisms lead to anatomical and functional changes in the brain. Our review shows that there is no clear evidence of a positive short-term effect of IF on cognition in healthy subjects. Clinical studies show benefits of IF for epilepsy, Alzheimer’s disease, and multiple sclerosis on disease symptoms and progress. Findings from animal studies show mechanisms by which Parkinson’s disease, ischemic stroke, autism spectrum disorder, and mood and anxiety disorders could benefit from IF. Future research should disentangle whether positive effects of IF hold true regardless of age or the presence of obesity. Moreover, variations in fasting patterns, total caloric intake, and intake of specific nutrients may be relevant components of IF success. Longitudinal studies and randomized clinical trials (RCTs) will provide a window into the long-term effects of IF on the development and progress of brain-related diseases.

## 1. Introduction

Brain diseases are among the leading causes of death and disability worldwide, becoming more important as the incidence has grown substantially in the past decades [1,2]. Despite the large number of studies that have been initiated to find possible treatments of brain-related diseases, therapeutic options are still mostly based on symptom relief while cures have not yet been found. Epidemiological evidence supports a role for life style factors that can open new potential avenues to aid the prevention of brain-related diseases [3]. For instance, the interplay between diets and their effect on the brain [4]. Several diets are found to support brain health, with most evidence pointing toward the Mediterranean diet (for a review see Scarmeas et al. [5]), which is high in vegetables, fruits, legumes, nuts, beans, cereals, grains, fish, and olive oil [6]. Moreover, the Dietary Approaches to Stop Hypertension (DASH) diet is designed to reduce cardiovascular risk, consisting of foods low in sodium, potassium, magnesium, and calcium, which decreases blood pressure and therefore the risks of stroke, dementia, and neurocognitive dysfunction [7]. The Mediterranean-DASH Intervention for Neurodegenerative Delay (MIND) diet is a combination of the Mediterranean diet and the DASH diet, specifically composed of food that are associated with slowing down cognitive decline [8]. Diets that involve caloric restriction have also shown positive effects on resistance against cognitive decline (for an overview of clinical improvements see Yu and colleagues [9]).

Although diets and calorie restriction have positive effects of brain health [10], such strategies are notoriously hard to sustain over time for many people and can have detrimental effects on people who already have low body weights or muscle mass [11,12]. Interestingly, a growing body of evidence from animal- and human (observational and clinical) studies suggest that fasting periods without caloric or nutritional changes could have similar effects on cognition and brain health [13]. So, alongside the growing interest in examining the role of nutritional intake on cognition, there has also been a growing interest to examine the timing and frequency of when to eat [14], called intermittent fasting (IF; see Section 1.1).

### 1.1. Different Variants of IF

Three variants of IF can be distinguished (see Figure 1), namely time-restricted eating (TRE), which is called time-restricted feeding in animals (TRF), alternate day fasting (ADF), and the 5:2 diet or periodic fasting (PF). ADF entails that people alternate between eating regularly on one day and restrain from eating the next day. PF is characterized by cycles of abstinence or strong limitation of food for 2 days a week whereas food can be eaten without restrictions for the other 5 days of the week. TRF is characterized by a time window of food intake that only lasts 8 h per day (note that studies vary on this and that eating windows of 6–12 h per day is also seen as TRF). There is also a distinction within TRF, namely eTRF (eating early during the day) and lTRF (eating late during the day). Fasting Mimicking Diet (FMD) is a variation of PF in which 5 consecutive days of low-calorie intake is practiced once a month [15].

IF is the abstinence or strong limitation of calories for 12 to 48 h alternated with periods of regular food intake with no restrictions. From an evolutionary standpoint, alternations of food availability and scarcity have been normal for most humans throughout history and could be coped with storing food as fat [16]. As a result of periods of restricted food intake, the human body initiates a metabolic switch from glucose to stored lipids, which leads to a cascade of metabolic, cellular, and circadian changes that are associated with numerous health benefits in animal models and humans [17,18,19]. Periods of IF have not only been associated with weight- and metabolism-related diseases, but also with reducing the risk/prevalence of neurological diseases [20]. In this review article, we will look at the effects of IF through which metabolic, cellular, and circadian mechanisms lead to anatomical and functional changes in the brain. Furthermore, we will critically review the evidence from clinical and epidemiological studies by listing studies that used different age groups, patient groups, and different dietary restrictions to obtain the most complete overview of the possible benefits of an intermittent fasting diet on cognition and brain-related diseases.

## 2. Metabolic, Cellular, Circadian, and Gut Microbial Responses to IF

### 2.1. The Metabolic Switch

The different variants of IF differ in the duration of the fasting period and therefore in their effects on metabolic function [14]. However, what they all have in common is that when IF is sustained long enough, a process called “flipping the metabolic switch” (Figure 2) is initiated [21]. This process occurs around 12 to 36 h after the fasting period begins, and depends on the initial liver glycogen content, the composition of the preceding meal and an individual’s amount of energy expenditure during the fast [21]. Flipping the metabolic switch entails that the body switches from its preference to extract energy through the process of glycogenolysis (breakdown of glycogen into glucose) to lipolysis (the utilization of stored fat in the form of lipids from adipose tissue). Subsequently, released lipids are metabolized to free fatty acids (FFAs) and are while first being transformed into the intermediate stage Acetyl CoA through the process of β-oxidation transformed to the ketones β-hydroxybutyrate (BHB) and acetoacetate (AcAc) [13].

What makes these ketones particularly interesting for cognition is that they become the preferred fuel for the brain during fasting periods [22]. Namely, in addition to the role of ketones as an energy source, these also regulate transcription factors (for example, CREB or PGC1α) in neurons [23]. BHB and AcAc are transported from the liver to the brain where they are metabolized back to acetyl CoA and HMG-CoA, which results in the upregulation of brain-derived neurotrophic factors (BDNF) [13]. The upregulation of BDNF is associated with the promotion of mitochondrial biogenesis, synaptic plasticity and cellular stress resistance in animal models [13]. Enhanced BDNF levels during IF are also found in humans [24] and it is hypothesized that enhanced circulating BDNF also leads to an increase in BDNF in the brain [13]. In animal models, the lowered levels of glucose during IF also leads to a reduction in the ATP:AMP ratio in neurons, which after some hours of fasting activates the AMPK and CaKMII kinases [25,26]. Activation of their downstream transcription factors (CREB and PGC1α) enables these kinases to inhibit anabolic processes, thus inhibiting cell growth and protein biosynthesis [25,26]. This, in turn, triggers repair by stimulating autophagy, a process where neurons remove dysfunctional or damaged components [27,28].

Neurons are able to regulate the synthesis of proteins in response to fluctuations in the availability of nutrition, namely through the mTOR pathway [29]. In a non-fasting state, activation of the mTOR pathway leads to protein and lipid synthesis. In contrast, activity of the mTOR pathway decreases during fasting periods and this leads to global inhibition of protein synthesis and the recycling of dysfunctional proteins by autophagy [30]. Autophagy is also responsible for the body’s ability to cope with oxidative stress (the accumulation of harmful free radicals) which deteriorates by age and during the progress of neurodegenerative diseases [31,32]. Inhibition of the mTOR pathway leads to an improvement in antioxidant defenses (molecules that prevent the oxidation of free radicals), DNA repair, and stimulation of BDNF [33]. Moreover, IF reduces inflammation, the body’s overreaction to injury or infection, through a reduction of monocytes in the blood, white blood cells that cause inflammation [34].

Intermittent fasting might also have an indirect beneficial effect on the brain through improvement of the insulin sensitivity [13]. Insulin sensitivity, the process of glucose cell absorption, is decreased in diabetic patients, but also naturally decays with age [35]. IF leads to decreased levels of circulating insulin in the blood, which enhances the sensitivity of insulin receptors and upregulates the insulin/IGF-1 signaling (IIS) pathway [36], leading to enhanced uptake and utilization of glucose by neurons [13]. Upregulated IIS activity also decreases the activity of the mTOR pathway [37] and is associated with enhancement of neuroplasticity and protection against oxidative stress [36].

### 2.2. Circadian Clock Mechanisms

Organisms have evolved to optimize physiological processes, such as the hormonal secretion pattern, to an endogenous circadian clock that matches day and night oscillations [38]. In humans, the brain area involved in regulating this circadian clock is the suprachiasmatic nucleus (SCN), which is entrained to light and dark. On a molecular level, the circadian clock is regulated by transcription factors that when rising too strongly inhibit their own expression through transcriptional-translational feedback loops [17]. Specifically, the transcription activators (BMAL1 and CLOCK) bind to three Period (Per 1–3) and two Cryptochrome (Cry 1–2) genes, driving their transcription. The translated proteins PER and CRY then inhibit the expression of BMAL1 and CLOCK, thus inhibiting their own expression. This creates a negative feedback loop in which gene transcription, hormonal secretion and protein levels oscillate on a ~24 h basis. The amplitude (the difference in levels of hormones, proteins, etc. between peaks and throughs) should be as large as possible to optimally prepare the body for activity or rest [39].

Similar secondary clock oscillators have been found in peripheral tissues, such as the liver, with meal timing as the main regulator [40]. Ideally, central and peripheral oscillators act in synchrony to optimally prepare the body for rest or activity. In western societies, 24 h lighting, shift work, and altered meal schedules lead to different input signals to the central and peripheral clocks [41]. For example, consuming food outside the normal eating phase (i.e., late-night eating) may set some peripheral clocks out of phase with central oscillators and dampens the amplitude. The amplitude of peripheral circadian oscillations increases with age [39] and is altered e.g., in ADHD [42], mood disorders [43], Alzheimer’s disease [44], hemorrhagic stroke vulnerability [45].

Interestingly, the IF variants PF and ADF have the ability to reset secondary oscillators which can be beneficial when central and peripheral oscillators are regularly out of phase. IF variant TRF has the ability to shift peripheral oscillators to match the phase of central oscillations. One way by which IF has effects on circadian rhythmicity is through hormonal synchrony. The peak of the circadian insulin secretion is reached in the early morning, which is further augmented during and after food intake [46,47]. Everyday practice of TRF in the morning decreases both post-meal and mean 24 h insulin levels, leading to an overall increased insulin sensitivity [24,48]. This is beneficial since glucose metabolism rates decay with age and are associated with Alzheimer’s disease, even before the onset of the disease [49,50].

Another mechanism by which IF variant TRF can alter circadian-driven processes is the result of downstream effects of the inhibited mTOR pathway [17,51]. During fasting periods, the expression of CRY1 and CRY2 is directly regulated through mTOR phosphorylated kinases (AMPK, CK1, and GSK3). Similarly, the mTOR pathway also increases the circadian phosphorylation of CREB which can activate Per transcription [52]. Through these mechanisms, practice of TRF affects circadian rhythmicity, which can lead to coupled and strengthened peripheral and central gene-, hormone-, and protein secretion. Therefore, TRF leads to optimal rhythms of behavior, physiology, and metabolism and ensures that anabolic and catabolic types of mechanisms are regulated in harmony with someone’s activity and rest cycle [17].

### 2.3. Gut Microbiota and the Gut-Brain Axis

An interesting mechanism mediating the effect of IF on brain health and cognition is the microbiota-gut-brain axis (MGBA). The human gastrointestinal tract is colonized by trillions of microorganisms or gut microbiota, collectively termed the gut microbiome. A higher diversity (richness) of microbiota is associated with healthier metabolic markers such as increased insulin sensitivity [53]. The composition of the gut microbiome is particularly interesting for cognition and brain-related disorders because there is increasing evidence that the composition of the gut microbiome directly influences the brain through neural, endocrine, and immune pathways, collectively called the microbiota-gut-brain axis [54,55]. The MGBA has several modes of action through which the gut microbiota can affect the brain. First, the microbiota modulates the interaction between the enteric nervous system and the central nervous system through the vagus nerve [56]. Second, the gut microbiota produces microbial (neuro) metabolites, signaling molecules which exert their effect by functioning as substrates for metabolic reactions. Third, the gut microbiota also has an indirect effect on the brain and behavior through the effects on immune system activation [54].

The diversity in gut microbiota composition depends on several factors, of which diet is a major one as well as dietary timing [57]. The abundance of ±15% of microbiota dynamically oscillates in activity and relative abundance throughout the day in response to circadian and hormonal fluctuations and moments of dietary intake [58,59]. Microbiota play a role in processes like the digestion of food components, host metabolism, and the maturation and function of the immune system, all of which show some degree of circadian control [60,61,62]. It is hypothesized that dynamically oscillating microbiota respond to and accommodate diurnal fluctuations in the environment such as feeding timing [58]. A western diet, eating close to or during the rest period, dampens microbiota oscillations [20,63], leading to a less diverse gut microbiome [59]. Interestingly, TRF is able to restore these cyclic fluctuations and thereby contribute to a richer diversity of the gut microbiome, even when nutritional intake is unaltered [59].

The gut microbiota may, through their role in metabolism, circadian rhythms, and immune functioning, mediate the effects of IF on brain health and cognition. Several animal studies have indeed found that IF changes the composition of the gut microbiota [19,64,65]. In the study of Liu and colleagues, IF enriched the gut microbiome composition and altered microbial metabolites which led to improved cognitive functioning, for example in spatial memory tasks [19]. Antibiotics treatment, detrimental for the gut microbiota, suppressed this improvement [19]. In a study of 80 healthy men, Zeb and colleagues found that TRF enriched the composition of gut microbiota, which led to up-regulated transcription of the Bmal1 and Clock genes and thereby improved circadian oscillations [66]. The gut microbiota is involved in the pathogenesis of various central nervous system disorders in humans like Alzheimer’s disease, Parkinson’s disease, epilepsy, and multiple sclerosis [67,68,69,70]. A (pilot) clinical study that examined the role of the gut microbiota in multiple sclerosis found that changes in gut microbial phyla were similar between mice and humans, though this was based on a very low samples size (*n* = 5 for control and IF group) [64].

### 2.4. Summary of IF Mechanisms Relevant for Brain Health and Cognitive Functioning

There are multiple ways by which IF can have effects on the brain (i.e., through the vagus nerve, (neuro) metabolites or immune activity). The human body initiates a metabolic switch from glucose to stored lipids after a period of restricted food intake. These lipids are metabolized to ketones, which have signaling effects and regulate transcription factors in neurons in the brain. Anabolic processes are minimized (such as protein synthesis and growth) and catabolic processes are favored that enhance stress resistance, tissue repair, and recycling of damaged proteins and molecules. Moreover, IF ensures that anabolic and catabolic types of mechanisms are regulated in harmony with individual cycles of activity and rest. Namely, IF has the ability to strengthen the amplitude and change the phase of secondary oscillators to match central oscillations of the SCN. Finally, IF enriches the diversity of the gut microbiome, which through the microbiota-gut-brain axis, leads to anatomical and functional changes in the brain. All in all, metabolic, cellular, and circadian mechanisms of fasting periods have direct and indirect influences on the brain which subsequently could improve cognitive functioning and the prevention or progression of brain-related disorders.

## 3. The Effect of IF on Brain-Related Disorders

Neurological diseases are major causes of morbidity throughout the world [2]. Neurodevelopmental and psychiatric disorders can cause long-lasting personal, social, and emotional difficulties [71]. Because of the aforementioned metabolic, cellular, and circadian effects when fasting, IF may have great potential to treat/prevent brain-related diseases. In general, available data on the direct effects of IF on mechanisms contributing to the development of brain-related diseases in humans are scarce. However, potential efficacy of IF on brain-related diseases in humans can be deducted by comparing IF-related protein and gene alterations in fasting humans to those in fasting animals. Namely, it is possible to measure and compare gut microbiota, signaling proteins, and gene expression during IF in both humans and animals. But most importantly, when available, randomized controlled trials (RCTs) will give the best insight in the possible positive effects of IF on brain-related diseases in humans.

The findings from preclinical and clinical studies on IF in brain-related disorders are summarized in the following section (see Table 1). While the focus of this review is on IF, a ketosis-inducing diet (ketogenic diet) is believed to have similar effects as IF [72] due to its low carbohydrate intake. Therefore, and also given the scarcity of IF studies, ketogenic diet was included in the search term and relevant results are summarized in Section 3.4.1. A comprehensive search of the electronic databases PubMed and Google Scholar for peer-reviewed articles published in English was conducted in the first week of January 2021 and updated in August 2021. Search terms were “Intermittent fasting”, “IF”, “Alternate day fasting”, “ADF”, “Time-restricted feeding”, “Time-restricted eating”, “TRF”, “TRE”, “Fasting mimicking diet”, “FMD”, “Ketogenic diet” (group 1) and “Alzheimer’s disease”, “Parkinson’s disease”, “Multiple sclerosis”, “MS”, “Ischaemic stroke”, “Epilepsy”, “Autism spectrum disorder”, “ASD”, “Mood disorder”, “Anxiety disorder”, “Depression” (group 2). These terms were combined as follows: group 1 and group 2. Again, due to the scarcity of experimental work and width of the topic of this review, a systematic review or meta-analysis is not possible. The studies found in these searches are hence summarized in a narrative review providing an overview of the advances of IF in brain-related disorders and cognitive functioning.

### 3.1. IF and Neurodegenerative Diseases

#### 3.1.1. Alzheimer’s Disease

The underlying mechanism that causes Alzheimer’s disease (AD) is unknown. It is known, however, that AD is pathologically characterized by beta-amyloid (Aβ) plaques and neurofibrillary tangles, leading to neuronal death, which is clinically characterized by a decay in cognitive abilities. Several studies using animal models have indicated that IF could reduce the accumulation of Aβ plaques and slow down cognitive decline [89,90,91]. Since the exact mechanism of AD is not yet fully understood, the mechanisms by which IF can have effects on AD is also only open for speculation. It is argued that IF can decrease and/or prevent AD-related neuropathology and cognitive decline by upregulating neuronal stress-resistance pathways and suppress inflammatory processes through decreased activity of the mTOR pathway [13] (see also Section 5.1 on prevention of age-related neurological disorders and cognitive decline). In the brain, there is a reduction in glucose metabolism rates with age, which can be present long before the onset of AD and is associated with Aβ plaque density [49,50]. Ketones may present an alternative energy source in a hypometabolic state [49,92,93]. In patients suffering from AD or mild cognitive impairment, injected BHB (a ketone) after approximately 12 to 16 h of fasting has led to improved cognitive functioning, assessed in various neuropsychological tests administered 90 min after injection [73]. In terms of IF, a 14-h TRF diet for 30 consecutive days has shown to reduce amyloid precursor protein (APP), the precursor of Aβ, in the blood of fourteen healthy subjects [74]. Ooi and colleagues found that a 3-year PF diet enhanced cognitive functioning in older adults with mild cognitive impairment compared to age-matched adults who irregularly practice PF and age-matched adults who do not practice PF [75].

#### 3.1.2. Parkinson’s Disease

Parkinson’s disease (PD) is characterized by the presence of α-synuclein-containing Lewy bodies and the loss of dopaminergic neurons in the substantia nigra (SN), which is clinically manifested by motor control problems (i.e., rigidity, bradykinesia, and tremor) and cognitive deficiencies [94,95,96]. An animal model of PD, in which the degeneration of nigrostriatal neurons causes PD-like behavior, can be induced by the administration of mitochondrial toxins that accumulate in dopaminergic neurons [13]. Using this model, neurotoxic-induced PD mice on a FMD showed greater retention of motor skills and less dopaminergic neuronal loss in the SN [76]. Specifically, a FMD reshaped the composition of the gut microbiota which through the signaling effects of metabolites restored the balance of astrocytes and microglia in the SN which are believed to be responsible for the inflammatory reactions in PD [76]. BDNF, important for the survival of dopaminergic neurons [97,98], was enhanced in mice practicing a FMD and was therefore speculated to have a role in the FMD-mediated neuroprotection [76]. Higher levels of BDNF were also found in macaque monkeys on a TRF regimen who were neurotoxically injected to mimic PD, which led to reduced motor deficiencies and attenuated dopamine depletion [77]. In humans, no clinical trials are yet initiated early in the disease process and continued long enough (1 year or longer) to detect a disease-modifying effect of IF.

#### 3.1.3. Multiple Sclerosis

Multiple sclerosis (MS) is an autoimmune disorder in which abnormal T-cell mediated inflammatory response of the body causes demyelination and axonal damage, leading to neuronal death [70,99]. Clinically, patients with MS show deficits in complex attention, efficiency of information processing, executive functioning, processing speed, and long-term memory [100]. MS is more common in Western countries with nutrition being a potential contributing factor, which led researchers to examine the role between IF and MS [64,70]. Three cycles of a FMD completely reversed disease progression in MS-induced mice [70]. A possible mechanism of IF on MS disability might be modulation of the gut microbiota, as 4 weeks of ADF activated microbial metabolic pathways and increased gut microbiota richness in a MS animal model [64]. This, in turn, led to lowered levels of T-lymphocytes, which are believed to be causative of MS pathogenesis [101]. Interestingly, transplantation of gut microbiota of MS-mice on an IF diet reduced MS pathogenesis for MS-mice without an IF diet [64]. In humans, a 7-day cycle of FMD led to lowered self-reports of MS disability in 60 MS patients [70]. In a small RCT with 5 MS patients and 9 controls, a 15-day ADF induced changes in the gut microbiota that are similar to what was observed in mice [64].

### 3.2. IF and Acute Central Nervous System Injury

#### 3.2.1. Ischaemic Stroke

Ischaemic stroke is characterized by a blockage of blood flow to a part of the brain leading to neuronal death and loss of (cognitive) functionality [102]. In animal models of focal ischemic stroke, rodents on a 3-month ADF diet prior to cerebral vessel occlusion exhibited reduced cortical neuronal loss and reduced cognitive decline in comparison with animals fed ad libitum [78]. Same results were obtained for the recovery of spatial memory deficits in rats maintained on a 3-month TRF diet before cerebral vessel occlusion compared with rats fed ad libitum [79]. During an ischemic attack, quick reperfusion of blood flow is associated with better clinical outcomes, but reperfusion is contradictorily associated with exacerbation of tissue injury [103]. Reactive oxygen species (ROS), a type of free radicals, have a critical role in initiating cell death and therefore enlarge tissue injury. Enhanced levels of ketones during a fast are thought to mediate the excitoprotective effects of IF by decreasing the levels of ROS [104]. Injected ketones after cerebral vessel occlusion in rats were found to decrease levels of ROS, which led to enhanced stress resistance as well as suppression of neuroinflammation, which are both positive for cell survival [105,106,107]. Interestingly, fasting initiated just after injury and maintained for 24 h reduced neuronal loss in rats [72], which could be clinically relevant for humans but up to this date has not yet been tested in clinical or randomized controlled trials. However, the Ulsan University Hospital of South Korea is currently examining the efficacy of TRF in a RCT by randomly assigning ischemic stroke patients to a 6-months TRF group or a control group [108]. In an observational study, Bener and colleagues reviewed the number of ischemic stroke hospitalizations for Muslims while fasting during the Ramadan (which is a type of TRF) and compared this incidence to non-fasting months [80]. However, they found no differences in the number of hospitalizations for stroke between Ramadan and non-fasting months.

#### 3.2.2. Epilepsy

Epilepsy is a neurological disorder characterized by recurrent bursts of abnormal excessive neuronal activity, named seizures, in which motor control and often consciousness is lost [109]. There is accumulating evidence that metabolic and biochemical effects of IF, including reduced blood glucose levels, inhibition of mTOR signaling, decreased inflammatory markers, increased AMPK signaling, and increased autophagy, leading to antiseizure and antiepileptogenic effects in animal models [110]. In an animal model of epilepsy, rats maintained on ADF for several months exhibited less neuronal hippocampal damage and showed improved performance on a spatial water maze after being induced with a seizure compared to seizure-induced rats fed ad libitum [81]. Similar results have been found for 7–10 weeks and 6 months of ADF in epilepsy-induced rats [111,112]. In children with epilepsy not responding to antiepileptic treatment, a PF regimen for two months improved seizure control in four out six children [82].

### 3.3. IF and Neurodevelopmental Disorders

ASD is a neurodevelopmental disorder which is clinically manifested by deficiencies in social communication and language, anxiety and repetitive behaviors [113]. Moreover, gastrointestinal symptoms are common comorbid symptoms in children with ASD [114]. It is hypothesized that alterations in (the early development) of the gut microbiome might be part of an ASD phenotype and increased risk of developing ASD [115]. Hence, dietary interventions such as IF could modulate ASD-like behavior. Limited initial evidence in this regard comes from an ASD mice model with a *Pten* haploinsufficiency (a gene associated with ASD in humans [116]) where the IF subtype ADF restored fear conditioning [83]. IF could also potentially affect ASD-like behavior through gut microbiome alterations, potentially increasing BDNF and ketone levels or by increasing mTOR pathway activity. For example, in an animal ASD model mTOR activity was reduced by amino acid dietary interventions [117,118,119]. Future clinical studies are necessary to see whether there are beneficial effects of IF on ASD symptomatology.

Research on the effects of IF on attention deficit hyperactivity disorder (ADHD) seems to be absent, see Section 3.4.1 for some work on the ketogenic diet in ADHD (and ASD).

### 3.4. IF and Neuropsychiatric Disorders

Mood- and anxiety disorders comprise a group of disorders that share a key feature of a general distorted emotional state, leading to feelings of sadness or anxiety, which clinically manifests in ensuing behavioral, emotional, cognitive, and physiologic responses [120]. We review the effects of IF on mood- and anxiety disorders together as there is high co-morbidity between these disorders [121]. BDNF levels, that are associated with both chronic stress [122] and chronic depression [123], were increased by a 9 h fast in a recent mice study also inducing antidepressant effects [84]. These effects of fasting were reversed by a 5-HT_2a_ receptor agonist, showing a link between fasting and this mood-related neurotransmitter system. In healthy humans, 6 months of IF improved mood as measured with the Hospital Anxiety and Depression Scale and World Health Organization Well-being Index [124]. Three months fasting in combination with calorie restriction in aged men reduced emotional reactivity symptoms such as tension and anger on the Profile of Mood States questionnaire, but not depression symptoms [125]. Moro et al. [85] found that TRF lowers inflammatory markers TNFα, Il-6, and IL-1b in 34 healthy subjects, that are associated with anxiety- and depression-like behavior [126,127], yet Moro et al. [85] did not measure effects of IF on mood. IF in psychiatric patients has to our knowledge only been examined in the context of Ramadan IF. Farooq et al. [86] found that Ramadan IF lowered the subjective feelings of depression and mania in 62 patients suffering from bipolar affective disorder. However, other studies highlighted the potential of relapse in bipolar disorder patients during Ramadan IF [87] or worsened schizophrenia symptoms [88].

#### 3.4.1. The Ketogenic Diet

While the ketogenic diet (which is high in fats and low in carbohydrates) is not a form of IF, its low carbohydrate composition can lead to similar effects as IF [110], given that the ketogenic diet leads to a rise in circulating ketone bodies [128]. For this reason, it is relevant to describe effects of the ketogenic diet in the context of brain disorders.

In the case of AD, ketones might be an alternative sources of fuel for the brain, as ketone uptake in the brain is not different for AD patients compared to healthy age-matched controls [128,129,130,131]. In mice, ketogenic diet counteracted AD pathogenesis and cognitive decline, indicated by improved performance on learning and memory tests and decreased Aβ and tau pathologies [132]. 

Mice models on epilepsy have shown that the ketogenic diet elevated GABA and decreased glutamate levels in the hippocampus, leading to less overexcitability in the hippocampus and fewer seizures [133]. Moreover, the gut microbiota mediated ketogenic diet-mediated seizure protection. Clinically, the ketogenic diet is implemented in medication-resistant childhood epilepsy, despite gastro-intestinal side effects in some patients [134]. Seizure relief in adult epilepsy is less studied and less established, as well as effects on cognitive functioning.

In terms of neurodevelopmental disorders, more studies examined the effects of a ketogenic diet on e.g., ASD compared to IF. A 3-week ketogenic diet increased social behaviors in an ASD mice model [135]. Here, the gut microbiota seems to act as a mediator between the ketogenic diet and its effect on ASD behaviors [136]. For example in an ASD mice model, a 10–14 day ketogenic diet elevated the fecal and cecal levels of *Akkermansia municiphila*, a bacterium important for a healthy intestinal wall [136]. In humans, two clinical studies showed reduced ASD symptoms after a ketogenic diet of 3 and 6 months in 15 and 45 children, respectively [137,138]. For ADHD, Packer and colleagues [123] observed reduced ADHD-related behaviors in an RCT of 21 dogs on a 6 months ketogenic diet, while measuring a significant increase in the ketone BHB. Human work on the effects of ketogenic diet in ADHD is lacking. 

Lastly, depression-like symptoms, measured in a rodent model, were reduced after elevated ketone levels induced by a ketogenic diet [139].

## 4. IF and Direct Effects on Cognition in Neurotypical People

An outstanding question is whether IF has positive effects on cognition for healthy subjects, not affected (yet) by psychiatric or neurological diseases. So far, no convincing direct effects of IF on cognition in healthy adults has been found. Benau and colleagues performed a systematic review on 10 studies wherein the effects of IF on cognition in healthy adults were examined when food intake was aligned with subjects’ regular eating pattern [140]. These studies showed an inconsistent profile, with either no changes due to IF or negative effects on executive function, psychomotor speed, or mental rotation. Given that young adults (aged 18–28) were tested here, potentially healthy adults do not benefit from IF due to a ceiling effect in their cognitive test results. Moreover, the studies mentioned in the review of Benau et al. have in common that subjects who do not regularly practice IF have to suddenly change to a fasting regimen [140]. This period is long enough to flip the metabolic switch, but too short to couple peripheral and central oscillators and therefore is often accompanied by sensations of hunger [141]. Hunger is associated with a decrease in cognitive performance [142] and could therefore be counter-effective for the effects of IF in short-duration trials. Namely, the subjective experience of hunger during fasting periods decreases as someone has regular fasts for a prolonged time [143].

Cognitive functioning has been researched in participants of fasting during Ramadan [144,145,146]. Qasrawi and colleagues reviewed studies that examined cognitive functioning during Ramadan IF and reported mixed results for psychomotor functioning, memory, and visual- and verbal learning with poorer performances observed later in the day [145]. Harder-Lauridsen and colleagues [147] found no change in cognitive functioning after 28 days of Ramadan IF. An important confound for Ramadan IF is that it partially reverses the normal circadian pattern of eating and drinking with the circadian clock regulated by day light. As mentioned before in this review, it is known that desynchronized circadian rhythms have detrimental effects on cognition [44]. Therefore, we will not go further into effects of studies on Ramadan IF.

## 5. Prevention of Neurological Diseases

### 5.1. IF Initiated in Different Age Groups

In order to understand the long-term effects of IF on the prevention of neurological diseases, its effects should be followed longitudinally, starting before the onset of a disease. For the effects of IF on neurological diseases rather than preventive attempts before their onset, see Section 3 and Table 1. Research on prevention includes measures such as longevity/mortality, which indirectly covers prevention of life-threatening diseases. Moreover a reduced risk of cardiovascular events and metabolic age are outcomes closely linked to, and thus used to measure prevention of neurological diseases [148,149]. In rats, ADF was initiated when rats are young, leading to a life span nearly twice as long [150]. When ADF was initiated in middle age, the rats lived 30–40% longer than rats fed ad libitum [151]. Two non-human primate studies showed that a decrease in caloric intake (while imposing TRF as well) is effective in delaying neurological disease onset and mortality [152,153]. In contrast to what was found in rodents, IF initiated in early age in non-human primates might even be counter effective for delaying the onset of neurological disease and enhancing life span [154], while IF onset in adult or advanced age yields clear benefits for survival in non-human primates.

In humans, there has not yet been a study that directly compared the effects of IF on the prevention of neurological diseases by longitudinally following subjects that started IF at a different ages. Therefore, the effects of IF on the prevention of neurological are studied in terms of reducing risks factors for neurological diseases. In 30 healthy non-obese middle-ages subjects, four weeks of ADF reduced cardiovascular risks factors (including a reduced risk for developing stroke) based on among others a lower Framingham Risk Score and reduced heart rate [155]. In another group with middle-aged subjects who followed > 6 months of ADF, heart rate as well as their blood lipid panel (i.e., LDL, HDL and triglycerides) was reduced [155]. There are more studies that report indications of a reduced risk for neurological diseases due to IF in healthy people. For instance, lowered APP levels [74] enhanced hippocampal neurogenesis [156] and decreased mTOR pathway activity [24], which are all protective of developing AD [157]. An interesting translational effort was done by Brandhorst et al. [15], where middle aged mice (20 months) and 38 human adults (age 20–68 years) followed a FMD protocol. Mice improved metabolic and age related symptoms such as visceral fat levels, immune-senescence but also cognitive functioning such as memory and in a subset of older mice signs of neurogenesis were observed. In humans, a FMD RCT (consisting of 5 days low-calorie intake in a month for 3 months) led to decreased C-reactive protein serum levels, which is an inflammatory marker and in high levels a risk factor for ischemic stroke [15]. However, cognitive functioning has not been measured in the human trial. Hence, an interesting question to ask is whether IF might have substantial effects on the prevention of neurological diseases in healthy people when people maintain IF for a longer period of time.

It is intuitive to think that starting at a young age would lead to healthier metabolic markers at an old age, but metabolic functions only start to worsen from a certain age [35]. As mentioned before, IF restores circadian rhythmicity and leads to decreased levels of circulating insulin in the blood, enhancing the sensitivity of insulin receptors [17,36]. Circadian rhythmicity and glucose metabolism rates in the brain are known to decline with age in healthy adults [35,39]. Therefore, this could indicate that IF might primarily have positive effects on cognition later in life when insulin sensitivity and glucose metabolism decays. However, Kim and colleagues did not find any differences of improvement in neurogenesis-associated memory after a PF diet between healthy younger subjects (from 35 years old) and healthy older subjects (till 75 years old) [156]. In addition, Brandhorst and colleagues also did not find age differences on reduced risk factors for age-related diseases and stroke vulnerability in healthy subjects after a 3-month FMD compared to a control group [15]. These findings do not provide an answer to the question whether it is more beneficial for neurological disease protection to start IF at a young age in comparison with starting IF later in life. Longitudinal studies with different age groups might be able to resolve this question.

### 5.2. IF Initiated in Obese and Non-Obese People

A next question that needs to be examined is whether possible protective effects of IF on neurological diseases might lie in the specific subject group that most studies use; people with or at risk for obesity. IF is most often examined in the context of a weight-loss diet for overweight or obese subjects [158]. The effects of IF on neurological disease prevention might lie in improving metabolic markers such as insulin sensitivity [159], which is not or less applicable to non-obese subjects. First, obesity is associated with a greater risk of developing neurological diseases [160]. Second, epidemiological studies indicate that obesity is associated with reduced cognitive functioning and cognitive impairment in older age, regardless of the presence of neurological diseases [161]. Third, obesity is also a risk factor for developing cardiovascular disorders and type 2 diabetes [162], which contributes to the development of vascular dementia, AD, and stroke [163,164]. The prevention of neurological diseases by IF might mainly be the result of the effects IF has on weight loss [158] and insulin sensitivity [165] in obese subjects. Namely, these would then indirectly lead to improved cognitive functioning and neurological disease prevention. Three clinical trials reviewed in this study have solely looked at the protective effects of IF on neurological diseases in obese subjects [24,156,166], while two clinical trials have found similar protective effects on neurological diseases in healthy non-obese subjects [15,74]. No study has directly compared the effects of IF between obese and non-obese subjects yet. Future studies have to resolve the question whether IF is similarly protective for neurological diseases in both obese and non-obese subjects.

## 6. IF vs. Other Dietary Interventions

### 6.1. IF vs. Caloric Restriction

An important question to address is whether (positive) effects of IF on brain health and brain disorders are triggered by the proposed fasting-induced metabolic, cellular, and circadian responses or by a reduced caloric intake. That is, as people have less time to eat during the day/week when practicing IF they therefore often eat less calories, a potential confounder driving the effects of IF. This is particularly relevant as caloric restriction (CR) is associated with health and survival. For instance, rats fed a limited amount of food lived much longer than ad libitum-fed rats [167], which has been replicated in many different species [168]. In non-obese humans, similar results have been found in epidemiological studies of centenarians living in Okinawa, exposed to CR most of their lives [169] and in a 2-year clinical trial [170]. Besides longevity, improved verbal memory [171,172], executive function and global cognition [172], and working memory [173] are also observed after 3, 12, and 24 months of CR, respectively. CR even has signaling effects comparable to IF, such as upregulation of the IGF pathway, downregulation of the mTOR pathway, gut microbiota composition changes, and activation of AMPK and its effects on cell autophagy [174,175]. In many animal models CR is tested in an IF feeding schedule; food is only provided once a day and animals tend to eat all of their food as soon as it is made available [176]. As the reduction of caloric intake automatically leads to a longer fasting period, IF is a form of CR. To address the question whether IF is in itself beneficial rather than merely reducing caloric intake, Mitchell and colleagues compared life-span and disease onset for ad libitum-fed mice, mice on CR fed several meals a day, and CR mice fed once a day while keeping caloric intake similar to the mice fed multiple times a day [177]. Interestingly, the time spent fasting was directly related to the health- and life span extension for all mice in that study, with the single meal-fed mice living significantly longer and having delayed disease onset than the multiple-meal-fed mice. This would also explain why dietary dilution, a form of CR in which mice eat all day to compensate for the low energy in their food, does not lead to life-span extension [178,179]. In addition to life-span extension and delayed disease onset, IF also leads to improved preservation of cognitive-, sensory-, and motor function in IF-fed rodents compared to CR-fed rodents [180].

In humans, difficulties in distinguishing between (effects of) CR and IF poses similar problems, as people eat on average 25–33% less during ADF and PF (likely restricting caloric intake), which holds true regardless of gender, presence of obesity, or age [181,182,183]. Studies that keep the caloric intake of the control group similar to the ADF or the PF group are necessary to disentangle the specific benefits of IF from CR. TRE allows consumption of all required daily calories within a narrower time frame, resulting in a prolonged fasting period without a net reduction in calorie intake [184,185,186]. Keeping caloric intake similar between the TRF and control group is crucial, as observational studies have found that in non-supervised environments, subjects practicing TRF have lower caloric intake compared to baseline, ranging from ~200 kcal/day [147,187,188] to ~350 kcal/day [189]. Hence, comparing the effects of TRE versus a no-diet control group can also answer to the specific benefits of IF, but only when the amount of calories is similar in the TRF group and the control group. Jamshed and colleagues compared early-TRF (eating between 08:00 and 14:00) with a non-fasting control group, keeping the caloric and nutritional intake over the day exactly the same between both groups [24]. The researchers found that TRF, in contrast to the control group, led to an increase in the expression of several genes associated with autophagy (LC3A), the circadian clock (PER1, CRY1, CRY2, and BMAL1) and insulin sensitivity (SIRT1) [190]. Furthermore, they found an increase in BNDF and an increase in the expression of mTOR in the TRF group compared to the control group. In an ADF trial in humans by Stekovic et al. [155], many lipids and free fatty acids levels (polyunsaturated free fatty acids, alpha-tocopherol) were higher on fasting versus non-fasting days, whereas several amino acid levels were lower, including the amino acid methionine for which low levels are associated with longevity [191,192]. Hence, the mere act of fasting irrespective of CR alters lipid metabolism, according to Stekovic et al. [155] potentially due to lipolysis in adipose tissue and hepatic clearance of amino acids from circulation for glucogenesis.

### 6.2. Healthier Nutritional Intake during IF

Another question to ask is whether any possible effects of IF on the prevention and progress of neurological diseases further improves when the nutritional intake during the fed-period is also healthier. Diets like the Mediterranean diet, the MIND, and DASH diet prevent cognitive impairment [5] and lower the risk for neurological diseases like PD, AD, and ischemic stroke [193,194,195]. We found no studies directly comparing the effects of IF to these specific diets. However, the IF type FMD involves a healthier nutritional intake in itself. The FMD entails that people fast for 5 days a month, in which they eat a low protein/amino acid diet, rich in fat and complex carbohydrates [15]. In a 2-arm cross-over RCT comparing 3 months of FMD with an unrestricted diet, Wei et al. found a reduction in IGF-1 [196], which upregulates the IIS-pathway activity and leads to enhancement of neuroplasticity and protection against oxidative stress [36]. The nutrients in the FMD were selected based on their ability to lower IGF-1, reduce glucose and increase ketone bodies while maximizing nourishment and minimizing adverse effects [15]. In contrast, 6 months of IF [181] or 6 years of 20% CR [197] does not lead to a net reduction in IGF-1. Therefore, Wei and colleagues suggest that the observed reduction in IGF-1 is related to the low protein/amino acid content of the FMD [196].

There are reasons to expect larger health effects when the eating window of TRF is in the morning since insulin sensitivity, β-cell responsiveness, and the thermic effect of food are all higher in the morning than in the afternoon or evening [198,199,200]. Potentially this dietary timing may benefit circadian rhythms. Several human studies have found beneficial effects of eTRF, like increased insulin sensitivity, lowered blood pressure, oxidative stress and inflammation [48,85,201]. In contrast, lTRF, restricting food intake to the late afternoon or evening, had no or even negative effects [202,203,204]. However, there have not yet been clinical studies that directly compare the differences between a group on subjects on eTRF and lTRF. Additionally, there might be a difference in the effectiveness of TRF depending on the timing of specific nutritional intake [41,48]. For example, carbohydrate oxidation is highest during the morning [205], which means that the largest proportion of carbohydrates can best be consumed during the morning [41]. In individuals with type 2 diabetes, a high-energy breakfast and low-energy dinner increased GLP-1 levels throughout the day [206]. High levels of GLP-1 are associated with improved cognitive functioning, for instance in AD [207]. In contrast, a low-energy breakfast and high-energy dinner decreased GLP-1 levels. So besides fasting, timed and tailored nutritional intake during fed periods might also be of influence for cognitive functioning in neurological diseases like AD.

TRF itself may lead to a healthier eating pattern. For instance, the time window in which people can eat is narrowed during TRF. Snacks, low in nutrients but high in “empty calories”, are most frequently consumed during the evening [208]. In a study with 13 healthy participants, the TRF group consumed significantly less snacks compared to the control group, especially during the evening [209]. This could also be an effect of research participation itself, as subjects in diet and lifestyle research tend to live healthier regardless of the specific manipulation [210]. Finally, unhealthy snacks and drinks are more often consumed during social events which take place during the evening [208]. Stockman and colleagues have highlighted the fact that IF might lead to skipping social events, thus indirectly leading to healthier eating patterns [211].

## 7. Summary and Discussion

Here, we reviewed the effects of IF on (prevention of) brain-related disorders and cognitive functioning by giving an interdisciplinary overview of the preclinical and clinical studies in this field. A comprehensive overview can only be achieved when knowledge about mechanisms on the molecular and cellular level are combined with insights on the clinical and psychosocial effects of fasting. The metabolic, cellular, and circadian mechanisms by which IF can lead to structural and functional changes in the brain are well described in animal models. The gut microbiota are an important mediator between dietary timing and circadian mechanisms and immune functioning, hereby also affecting the central nervous system. While no clinical IF studies are performed on PD so far, ischemic stroke and mood- and anxiety disorders, animal models indicate a remodeling of the gut microbiome and reduced neuronal loss in PD, reduced neuronal loss and cognitive loss in ischemic stroke, and heightened BDNF levels inducing antidepressant-like effects in mood- and anxiety disorders.

In humans, the number of clinical studies examining the effects of IF on neurological diseases is still limited. In these, positive findings have been found for several neurological diseases; clinical trials show that different types of IF (TRF, PF, ADF, and FMD) improve seizure control in epilepsy, improved cognitive functioning in AD, and lowered self-reports of disability through enrichment of the gut microbiome in MS. Mood- and anxiety symptoms could benefit from IF though several potential postulated pathways, but properly sample sized, randomized, controlled clinical trials are needed to confirm this initial evidence. Research (both animal and clinical) on IF in neurodevelopmental disorders such as ASD and ADHD is in its infancy.

In healthy people IF does not lead to any short-term benefits for cognition and long-term clinical trials to examine the effects of IF on cognition or neurological disease protection have not yet been initiated. However, there are indications that IF might be protective of developing neurological disorders, as studies report a lowered risk for ischemic stroke or AD in healthy subjects [15,24,74,155,156]. The optimal starting point of IF across the lifespan for the prevention of neurological diseases has not been determined as most clinical studies examined subjects with a wide age-range [15,156] and longitudinal clinical studies are lacking. For example, beneficial effects of IF on enhancing insulin receptor sensitivity and improving circadian gene expression might be more efficient later in life when circadian rhythmicity and insulin sensitivity decay [35,39]. Animal studies looking into the starting point of IF have conflicting findings. A delayed positive effect is seen in rodents; rodents fasting from birth live significantly longer than those that start later in life [150,177]. In contrast, IF initiated at an early age in non-human primates is found to be counter effective in delaying the onset of neurological disease and enhancing life span [154]. Large longitudinal studies with different age groups are necessary to understand the long-term effects of IF in humans and whether these should be targeted especially at older adults with or at risk for neurological disorders.

A potential increased efficacy of IF on brain and cognitive functioning in an obese population has not been tested. IF leads to improvements in weight loss and insulin sensitivity, which indirectly might be protective of neurological disease prevention and progress [158,165]. Effects of IF on physical health may equally affect brain health in individuals with obesity and these may be larger compared to subjects with a healthier metabolic state. Future clinical studies should shed light on whether IF has a stronger effect on brain health in obesity.

At last, we examined whether IF is more effective for brain and cognitive functioning depending on the type, amount and timing of nutritional intake. There is variability in the way IF is practiced, yet relevant differences between (effects of) these variants are currently not sufficiently accounted for in the literature. As IF and CR trigger the same mechanisms [175], a caloric reduction during IF might further strengthen neurological health benefits. Specifically for obese subjects CR during IF might lead to more positive results since all types of IF (with the exception of TRF) lead to weight loss [211]. Intuitively, an IF diet combined with a diet composed of nutrients that promote brain health (e.g., the Mediterranean diet or the MIND diet) would be expected to give even more positive results, yet no studies directly examined this. There have been positive results for FMD, which involves fasting combined with a low protein/amino acid diet. Future studies should directly compare FMD with other types of IF to disentangle the specific contribution of nutrition. Efficacy of IF may also vary depending on the (healthy) content of the diet [15,39], whether the eating window is early or late during the day [48]. In addition, it might be that IF itself leads to a healthier eating pattern. Namely, the time window in which people can eat is narrowed during TRF, which indirectly might lead to a healthier eating pattern [208,211].

## 8. Conclusions

Animal studies show clear mechanisms by which IF has positive effects on brain-related disease models while clinical studies are mostly still at infancy. IF does not lead to any short-term benefits for cognition in healthy people, but there are indications that IF might be protective of developing neurological disorders. Future research should disentangle whether this protective effect holds true regardless of age, the presence of obesity, total caloric intake, and the timing and intake of specific nutrients (see Section 8.1). Lastly, while theoretically IF may be beneficial also for neurodevelopmental and mood disorders, hardly any experimental data exist on this topic.

### 8.1. Open Questions and Future Outlook

Future research should look into the question for whom and when IF yields positive effects on brain health and cognition. A first step is to use controlled paradigms and take into account how design choices and subject recruitment could influence results. 

First, different variants of IF (ADF, PF, TRF, and FMD) are all combined into the term IF. However, there are clear differences in the duration of the fast [12], the reduction in the amount of calories [191] and (mis)alignment with circadian rhythms [14] between the variants which can possibly influence cellular, metabolic, and circadian effects of IF. Second, while caloric restriction has different effects on longevity and cognition in animals when it is initiated at a different age [158,159,160,161,162], no studies have examined this in the context of IF. Longitudinal studies of IF, initiated with subjects at young and middle age and continued until an age where neurological diseases commence, could provide a window into the long-term effects of IF on the development of neurological disorders. Since this would be a time-consuming effort, the long-term effects of IF could also be understood by mechanistic studies examining risks factors for neurological diseases [23,74,163,164,165]. Third, a next question that needs to be examined is whether possible protective effects of IF on neurological diseases might lie in the specific subject group that most studies use; people with or at risk for obesity [166,167,168,169,170,171,172,173]. No study has directly compared the effects of IF between obese and non-obese subjects yet. Future studies have to resolve the question whether IF is similarly protective for neurological diseases in both obese and non-obese subjects. Next, studies should always control if observed effects are caused by a prolonged time of fasting, or through caloric reduction as a side effect of fasting. Most variants of IF lead to a net reduction in caloric intake [155,189,190], so studies should either control for caloric intake, or set fixed meals for experimental and control groups. Finally, the type and timing of nutritional intake might play an important role in the effectiveness of IF [205,207,208,209], which can also be overcome with fixed meals.

Besides controlled paradigms, little to no research has been performed on the effects of IF on neurodevelopmental and psychiatric disorders while animal models and ketogenic diets in humans show possible modes of action. It is also unknown whether positive effects of IF are the result of direct effects of the brain, or a general health improvement through insulin sensitivity [166] or weight loss [173]. Given the positive outcomes thus far, IF may prove to be a promising approach for improving brain health once it is determined which individuals will best benefit from it.

## Figures and Tables

**Figure 1 nutrients-13-03166-f001:**
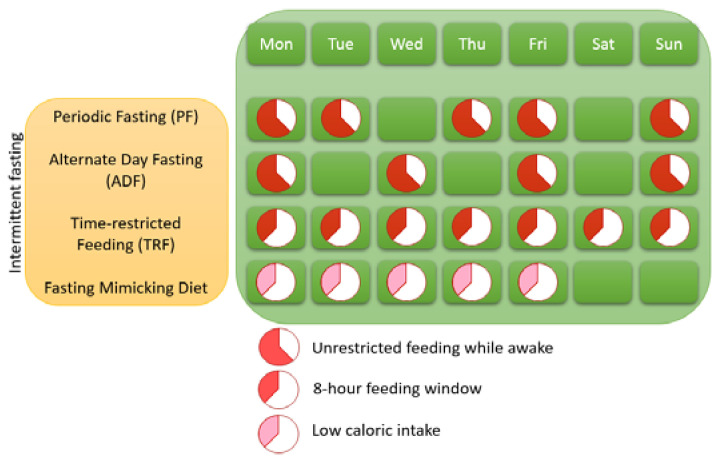
Different forms of intermittent fasting.

**Figure 2 nutrients-13-03166-f002:**
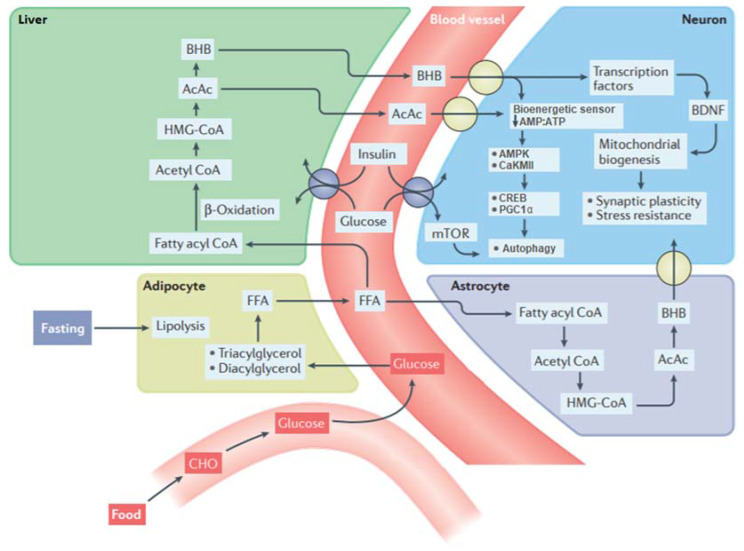
Biochemical pathways involved in the metabolic switch. During intermittent fasting, glucose levels drop and through the process of lipolysis, fats (triacylglycerols and diacylglycerols) are metabolized to free fatty acids (FFAs). These lipids are then transported to the liver where they through the process of β-oxidation and the intermediate stages acetyl CoA and HMG-CoA are transformed into the ketones: acetoacetate (AcAc) and β-hydroxybutyrate (BHB). BHB and AcAc are transported from the blood into the brain and then into neurons. In addition to ketones metabolized in the liver, astrocytes are also capable of ketogenesis, which may provide an important local source of BHB for neurons. The reduction in availability of glucose and elevation of ketones lowers the AMP: ATP ratio in neurons, which activates the kinases AMPK and CaKMII and, in turn, through the activation of CREB and PGC1α stimulates autophagy. In addition, lower levels of glucose during fasting decrease the activity of the mTOR pathway, leading to autophagy. BHB can also upregulate the expression of brain-derived neurotrophic factor (BDNF) and may thereby promote mitochondrial biogenesis, synaptic plasticity, and cellular stress resistance. IF leads to lower levels of circulating insulin in the blood, which enhances neuroplasticity and protection against metabolic and oxidative stress through the insulin/IGF signaling pathway. Retrieved from [18] with small modifications.

**Table 1 nutrients-13-03166-t001:** Characteristics of relevant preclinical and clinical studies on Alzheimer’s disease, Parkinson’s disease, multiple sclerosis, ischemic stroke, epilepsy, autism spectrum disorder, and mood- and anxiety disorders (see note below).

Brain-Related Disorder	Species	Type of IF	Duration	Reference	Findings
	Humans	Fasting	12–16 h	Reger et al. [73]	Injected ketones leads to improved cognitive functioning while fasting in patients with AD or MCI
	Humans	TRF	30 days	Mindikoglu et al. [74]	Reduced amyloid precursor protein in healthy subjects
	Humans	PF	3 years	Ooi et al. [75]	Enhanced cognitive functioning in MCI patients
Parkinson’s disease	Rodents	FMD	3 cycles	Zhou et al. [76]	Greater retention of motor skills and less dopaminergic neuronal loss in the substantia nigra (MPTP PD model)
	Macaques	TRF	6–10 months	Maswood et al. [77]	Reduced motor deficiencies and attenuated dopamine depletion (MPTP PD model)
Multiple sclerosis	Rodents	FMD	3 cycles	Choi et al. [70]	Reversed disease progression (EAE model)
	Rodents	ADF	4 weeks	Cignarella et al. [64]	Increased gut microbiota richness and lowered levels of T-lymphocytes (EAE model)
	Humans	FMD	7/30 days	Choi et al. [70]	Lowered self-reports of multiple sclerosis disability
	Humans	ADF	15 days	Cignarella et al. [64]	Reduced inflammation and enhanced protective changes of the gut microbiota
Ischaemic stroke	Rodents	ADF	3 months	Arumugam et al. [78]	Reduced cortical neuronal loss and reduced cognitive decline (stroke induced using cerebral artery occlusion)
	Rodents	ADF	3 months	Roberge et al. [79]	Recovery of spatial memory deficits (stroke induced using cerebral artery occlusion)
	Rodents	fasting	24 h	Davis et al. [72]	Reduced neuronal loss when fasting is initiated after moderate injury and maintained for 24 h
	Humans	Ramadan IF	13 years	Bener et al. [80]	No differences in the number of hospitalisations for stroke between Ramadan and non-fasting months assessed in an observational study
Epilepsy	Rodents	ADF	2–4 months	Bruce-Keller et al. [81]	Less neuronal hippocampal damage and improved spatial navigation (using excitotoxin kainate epilepsy model)
	Humans	PF	2 months	Hartman et al. [82]	Improved seizure control in children
Autism spectrum disorder	Rodents	ADF	60 days	Cabral-Costa et al. [83]	Rescued fear conditioning in ASD mice (*PTEN* haploinsufficiency ASD model)
Mood- and anxiety disorders	Rodents	fasting	9 h	Cui et al. [84]	Increased serotonin receptor dependent prefrontal BDNF and c-Fos levels and antidepressant effects (reduced immobility during forced swimming)
	Humans	TRF	8 weeks	Moro et al. [85]	Lowered inflammatory markers
	Humans	Ramadan IF	30 days	Farooq et al. [86]	Lowered subjective feelings of depression and mania
	Humans	Ramadan IF	30 days	Eddahby et al. [87]	Relapse in bipolar disorder
	Humans	Ramadan IF	30 days	Fawzi et al. [88]	Worsened schizophrenia symptoms

**Note:** The species on which the study is conducted, the type of IF, and the duration of the diet are shown in columns two to four. The references of the studies are shown in the fifth column. Main findings of each study are reported in the last column. ADF, alternate day fasting; PF, periodic fasting; TRF, time-restricted feeding; FMD, fasting-mimicking diet; AD, Alzheimer’s disease; MCI, mild cognitive impairment, MPTP; 1-methyl-4-phenyl-1,2,3,6-tetrathydropyridine; EAE, experimental autoimmune encephalomyelitis; PTEN, phosphatase and tensin homolog; ASD, autism spectrum disorder; BDNF, brain derived neurotropic factor.

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
