# Peer review of "The Effects of Intermittent Fasting on Brain and Cognitive Function"

_nutrients, 2021, doi:10.3390/nu13093166_

Round 1

Reviewer 1 Report

Dear Authors,

The article as a whole is very interesting and the literature review is done quite deeply. It presents many of the issues needed to understand the concept of IF, not only in the context of the brain and cognitive function but the response of different parts of the body to the effects of dietary restrictions. Despite the many positive values of the article, the authors did not shy from some errors, and some of the descriptions are not quite clearly presented.

Notes in lines and article:

39 - it was stated that most research evidence points to the Mediterranean diet, but only one citation was added.

40 - the effect of Dietary Approaches to Stop Hypertension (DASH) on supporting brain function is not clear, a citation is recommended.

50 - this is the result of a study from several years ago, there are no better ones?

126 - figure 2, is missing in the manuscript.

227 - citing article 69 here, may mistakenly suggest this was found after a human study, while it was an animal study.

229 - is written subsection 2.5 and should be 2.4.

229 - At the end of the subsections of Chapter 2, there are summaries of them. Is it then necessary to have another summary of these in a subsection, now called 2.5, where there is also a summary at the end? Similar summaries are found later in the work, it is worth considering whether they are justified and when to use them, they occur not always.

276 - Table 1. is worth starting on a new page. Also, wouldn't it be better to keep the titles of tables and figures short/informative?

Please note the splitting of text in Table 1, some of it is incorrect, like:

"Human

s"

Macaqu

es"

What does the study "University of Genova - Ongoing random clinical trial to assess the effectiveness of FMD in MCI patients", "Ulsan University hospital - Ongoing random clinical trial to assess the effects of TRF after stroke" bring to the table?

What is written about Alzheimer's, agreed.  Isn't Alzheimer's the result of a so-called "starved" brain? The problem occurs in the mitochondrion, which does not get food (glucose). It lacks food so the brain shortens the connections to keep the cell alive. Fuel can be ketones and glucose. With BDNF proteins it can be nourished and these are made during IF processes.

427 and 454 - why subsections 3.4.1. and 4.1. when there are no 3.4.2., 4.2.?

Subsection 5.1. is about neurological diseases. Please reflect on its content. It seems that it is not exhaustively and clearly described/explained, there is little on this topic and other topics are described, related to aging, or the vascular system. Part of the text is speculation.

The paper uses the abbreviation TRF, are you sure the abbreviation TRF is used correctly? What is the difference in the use of TRF versus TRE?

720 - 7.1. and there is no 7.2....Maybe you should create a separate section 8, or add conclusions to the description in line 665?

There are many valuable summaries and conclusions in the article. After reading the whole article, this thought occurred to me. Is it not the case that by using IF, organisms somehow recover the right to cure themselves? You just have to allow them to do so, and this can also be done through IF.

Author Response

Please see it in the attachment.

Reviewer 2 Report

The present manuscript from Gudden et al. will critically review the evidence from clinical and epidemiological studies by listing studies that used different age groups, patient groups and different dietary restrictions to obtain the most complete overview of the possible benefits of an intermittent fasting diet on cognition and brain-related diseases.

In Box 1 different variations of IF and the ketogenic diet were described in detail. In this box FMD is described as a variation of periodic fasting in which 5 consecutive days of low-calorie intake is practiced once a month - in citation #140 there is no information about FMD, please check!

Please check citation #76 - add the homepage or better the registration number of clinicaltrials.gov

Please add the citation-numbers in table 1, so it is easier to find the citation in the references

In my point of view neither FMD (fasting mimicking diet) nor ketogenic diet are forms of intermittent fasting. Therefore please exclude this studies from this analysis or describe it in the objective (aim of the analysis).  Fasting means no food intake. Both, FMD and ketogenic diet do not fit this definition.

Otherwise, the present work is a very clear, well-structured and investigated manuscript. The introduction is well written, the search terms for the comprehensive search are well chosen. The discussion is extensively and well cited.

Author Response

Please see it in the attachment.
